# Stronger Random Baselines for In-Context Learning

**Gregory Yauney & David Mimno**
Cornell University
{gyauney@cs.,mimno@}cornell.edu

## Abstract

Evaluating the in-context learning classification performance of language models poses challenges due to small dataset sizes, extensive prompt-selection using the validation set, and intentionally difficult tasks that lead to near-random performance. The standard random baseline—the expected accuracy of guessing labels uniformly at random—is stable when the evaluation set is used only once or when the dataset is large. We account for the common practice of validation set reuse and existing small datasets with a stronger random baseline: the expected maximum accuracy across multiple random classifiers. When choosing the best prompt demonstrations across six quantized language models applied to 16 BIG-bench Lite tasks, more than 20% of the few-shot results that exceed the standard baseline do not exceed this stronger random baseline. When held-out test sets are available, this stronger baseline is also a better predictor of held-out performance than the standard baseline, avoiding unnecessary test set evaluations. This maximum random baseline provides an easily calculated drop-in replacement for the standard baseline.

## 1 Introduction

One of the most exciting applications of contemporary large language models (LMs) is their ability to perform complex tasks given only a small number of examples (Brown et al., 2020). In-context learning (ICL), also called few-shot, performance has therefore become a critical tool in LM evaluation (Liu et al., 2023), but the nature of few-shot tasks makes them hard to contextualize. Because they are intended to evaluate specific abilities, datasets can be small and idiosyncratic. ICL performance is extremely sensitive to small changes in formatting and demonstrations (Zhao et al., 2021; Sclar et al., 2024). Finally, the fact that few-shot datasets are intended to evaluate the outer bounds of LM performance means that they are intentionally designed to be difficult (Suzgun et al., 2023). We study the implications of these characteristics and argue for using a probabilistic baseline that better distinguishes from random performance for small datasets and accounts for searches over prompts.

Downstream users want to find and deploy the best prompt (Mizrahi et al., 2024). But ICL performance of LMs varies greatly across semantically equivalent prompt features like the choice of demonstrations and their order (Zhao et al., 2021; Lu et al., 2022), instruction phrasing (Mizrahi et al., 2024), and template formatting (Sclar et al., 2024; Voronov et al., 2024). Because performance is both variable and unpredictable, standard ICL practice involves searching over large numbers of potential prompts based on a validation set. Researchers often report model performance as the maximum score on a validation set or the corresponding performance of that prompt on an additional truly held-out test set to avoid overfitting to the validation set (Brown et al., 2020; Perez et al., 2021). In this work we show that we can better identify prompts that may be overfitting and avoid unneccessary evaluations on test data.

In searching for the best prompts, the simplest and most common comparison is to a random baseline. The standard random baseline for classification tasks is the expected accuracy of guessing labels uniformly at random (Mitchell, 1997, *inter alia*). While universally treated as a point estimate, the accuracy of any *specific* random classifier follows a binomial distribution. For larger datasets, the variance of this distribution is tightly concentrated

around the expectation, but variance can be considerably higher for the small datasets typically used in the ICL setting. We introduce a stronger random baseline that accounts for both variance and validation set reuse by asking a fairer question: if we are choosing the best of $t$ different prompts, why not compare that prompt's accuracy to the best of $t$ different random classifiers? Figure 1 shows how these two random baselines can lead to different conclusions on a fixed evaluation set. We also find that the maximum baseline can provide a more accurate estimate of a prompt's generalization to a held-out test set than the standard baseline does. This can reduce premature usage of the test set.

Treating random performance as a distribution has two key advantages. First, the stronger random baseline can be calculated in closed form as the expectation of the maximum order statistic of the binomial distribution. When choosing the best prompt from even as few as 10 options, this baseline increases the threshold for beating "random" performance by more than 7 points of accuracy for a binary classification task with 100 examples. Second, by using a family of parametric distributions to represent random performance rather than a point estimate, we can also calculate a comparable performance metric across multiple datasets that takes into account factors like the number of classification options, the number of evaluated prompts, and the number of evaluated examples. We can then use a tail probability to quantify what fraction of random classifiers outperform a prompt. This contextualization also applies in the setting where evaluation data is used only once.

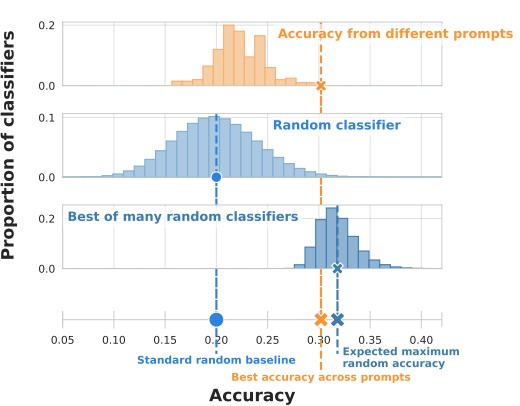

Figure 1: 200 different prompts for the emoji_movie task yield a spread of accuracies for OLMo-7B (4-shot, quantized). The best prompt has much higher accuracy than the expected performance of a single random classifier. But its performance is worse than the expected maximum accuracy among 200 different random classifiers.

Our contributions are the following. First, we propose a maximum random baseline that explicitly depends on validation set size and the number of evaluated prompts. We give a simple method for computing this value by taking the expectation of the maximum order statistic of the binomial distribution and show how this quantity varies with task parameters and evaluation setup. Second, we examine quantized LM few-shot performance across the BIG-bench Lite benchmark suite when choosing the best prompt demonstrations, finding in our setting that more than 20% of results that exceed the standard baseline do not exceed the maximum random baseline. Third, we show that when a truly held-out test set is available, comparing maximum validation performance to the maximum random baseline, rather than the standard random baseline, is better able to predict whether corresponding test performance will also be above random. This can help prevent prematurely evaluating with a held-out set, reducing the chance of overfitting to that held-out test set. The stronger baseline can also allow researchers to use smaller validation sets more confidently. Finally, the maximum random baseline can be easily calculated as a drop-in replacement for the standard random baseline.[1] High-quality evaluation datasets remain small, and prompt variability leads to dataset reuse. Our baselines should account for this.

## 2 Related work

**Robust model comparisons.** Dodge et al. (2019) introduce expected maximum validation accuracy to compare model accuracy for a given budget of hyperparameter evaluations. This value can be estimated with low mean-squared error (Dodge et al., 2021). Our work casts the number of validation set reuses as the resource of interest and asks a more basic question:

---

[1]Code is available at: https://github.com/gyauney/max-random-baseline

is a given model even outperforming random guessing for a set budget of validation set reuses? Other work also seeks to compare distributions of model performance (Dror et al., 2019), but it does not do so against a random baseline. Card et al. (2020) find that many common NLP evaluation datasets are too small to reliably compare models unless there is a large improvement in performance. Bragg et al. (2021) advocate for larger few-shot evaluations to reduce confidence intervals around reported performance. Our work finds one more reason to prefer larger evaluation sets.

**Overfitting to the evaluation set.** Data reuse has been approached from many angles. Some restrict access to the test set, either through submission systems (Wang et al., 2019a;b; Alex et al., 2021; Bragg et al., 2021) or differential privacy (Dwork et al., 2015). Perez et al. (2021) advocate for a *true few-shot* setting: model selection using cross-validation and minimum description length on the training set, with only one held-out set evaluation. While they compare selected prompts to randomly selected prompts, we instead compare to classifiers that guess at random. Some works create held-out sets for ICL (Wei et al., 2022; Nori et al., 2023), though this is not always standard practice. Rather than stipulate how to access the validation set, our approach contextualizes the current practice of reusing the validation set. Sometimes the prompt template with the best validation performance is chosen for test set evaluation (Wei et al., 2022; Mizrahi et al., 2024). The stronger random baseline does not apply to the test accuracy, but it does apply to the best validation accuracy.

**Permutation tests for classification.** Permutation tests have a long history for evaluating the strength of association between features and labels by retraining classifiers on datasets with shuffled labels (Golland et al., 2000; Ojala & Garriga, 2010), especially in low-data medical tasks (Golland & Fischl, 2003). In contrast, our approach seeks to determine improvement over a random *classifier* and—since the random performance distribution is known—we do not need to retrain any classifiers. Hypothesis tests have also been increasingly used in NLP to test whether one classifier reliably has better performance than another (Dror et al., 2018; Zmigrod et al., 2022; Peyrard et al., 2021).

## 3 Random baselines

Consider a dataset with $n$ validation examples and $m$ possible labels.[2] Then $p = 1/m$ is the probability of guessing one example's label correctly uniformly at random.[3] We primarily consider how to contextualize accuracy in a setting with just one set of $n$ examples (generalization is discussed in Section 5.2). An experiment evaluates $t$ different classifiers (or prompts, or hyperparameter settings) and reports maximum accuracy.

**Standard random baseline.** Let $h$ be a classifier that guesses labels uniformly at random for each example. Let $B(n, p)$ be the binomial distribution with $n$ independent trials and probability $p$ of success on each trial. Let $X$ be the number of correct guesses that $h$ makes when evaluated on all examples. $X \sim B(n, p)$ models the number of correct guesses, and:

$$\mathsf{acc}(h) = \frac{1}{n}X$$

The expected accuracy of a random classifier is straightforward:

$$\mathop{\mathbb{E}}_{h}\left[\mathsf{acc}(h)\right] = \mathop{\mathbb{E}}_{X \sim B(n,p)}\left[\frac{1}{n}X\right] = \frac{1}{n}(np) = p \tag{1}$$

**Expected maximum random baseline.** In this setup, we want a baseline comparable to the classifier that achieves the maximum accuracy on the validation set out of $t$ different classifiers. The idea is to take the expected maximum performance among $t$ random classifiers. Let $h_1, \dots, h_t$ be classifiers that guess answers independently and uniformly at

---

[2]Appendix A extends the baseline to datasets where $m$ varies per example.
[3]Appendix A also shows this does not depend on the dataset having balanced labels.

random, with corresponding independent numbers of correct guesses $X_1, \dots, X_t$. Consider the one with the highest performance:

$$h_{\max} = \underset{i \in [t]}{\operatorname{argmax}} \left\{ \operatorname{acc}(h_i) \right\}$$

The number of correct guesses $X_{\max}$ by $h_{\max}$ is the $t^{\text{th}}$ order statistic (or sample maximum) of the $X_i$, denoted $X_{(t)}$:

$$X_{\max} = \max_{i \in [t]} \left\{ X_i \right\} = X_{(t)} \tag{2}$$

Recall the probability mass function $f(k) = P(X_i = k)$ and distribution function $F(k) = P(X_i \leq k)$ for all binomial random variables $X_i$:

$$f(k) = \binom{n}{k} p^k (1-p)^{n-k} \qquad\qquad F(k) = I_{1-p}(n-k, 1+k) \tag{3}$$

where $I$ is the regularized incomplete beta function. Following the general method for order statistics of discrete random variables (Casella & Berger, 2002), the probability mass function for the $t^{\text{th}}$ order statistic, i.e. the sample maximum, is:

$$P\left(X_{(t)} = k\right) = P\left(X_{(t)} \leq k\right) - P\left(X_{(t)} < k\right) \tag{4}$$

$$= P(X_1 \leq k \wedge \dots \wedge X_t \leq k) - P(X_1 < k \wedge \dots \wedge X_t < k) \tag{5}$$

$$= P(X_1 \leq k)^t - P(X_1 < k)^t \tag{6}$$

$$= F(k)^t - \left(F(k) - f(k)\right)^t \tag{7}$$

We now have a closed form for the expected maximum accuracy out of $t$ random classifiers:

$$\operatorname{E}\left[\operatorname{acc}\left(h_{\max}\right)\right] = \underset{X_1,\dots,X_t}{\operatorname{E}}\left[\frac{1}{n} X_{\max}\right] = \frac{1}{n} \underset{X_1,\dots,X_t}{\operatorname{E}}\left[X_{(t)}\right] = \frac{1}{n} \sum_{k=0}^{n} k P\left(X_{(t)} = k\right) \tag{8}$$

$$= \frac{1}{n} \sum_{k=0}^{n} k \left(I_{1-p}(n-k, 1+k)^t - \left(I_{1-p}(n-k, 1+k) - \binom{n}{k} p^k (1-p)^{n-k}\right)^t\right) \tag{9}$$

**Tail probabilities against random baselines.**  For either kind of random baseline, it can be useful to compare to the *distribution* of random performances rather than just the expected accuracy. To get the probability that random classifiers outperform a given classifier $h_0$, the tail probabilities for the accuracy of $h_0$ against both baselines are (details in Appendix A):

$$p_{\text{standard}} = P(\operatorname{acc}(h) \geq \operatorname{acc}(h_0)) \qquad\qquad p_{\max} = P(\operatorname{acc}(h_{\max}) \geq \operatorname{acc}(h_0)) \tag{10}$$

$$= 1 - F(n\operatorname{acc}(h_0) - 1) \qquad\qquad = 1 - F(n\operatorname{acc}(h_0) - 1)^t \tag{11}$$

## 4 Properties of the expected maximum random baseline

This section builds intuitions for the values of maximum random baselines and how they are affected by experimental design parameters. The standard random baseline depends only on the number of possible labels. In contrast, the expected maximum random baseline depends on the number of validation examples $n$, the probability of guessing a label correctly $p$, and the number of validation set evaluations $t$. Figure 2 shows the expected maximum random accuracy on a binary classification task as a function of validation set evaluations $t$ and dataset size $n$.

**Expected maximum random accuracy is higher for smaller $n$ and larger $t$.**  For a given dataset size $n$, expected maximum random accuracy increases as $t$ increases. For a given number of validation set evaluations $t$, smaller datasets have greater expected maximum random accuracies. When there are fewer than several hundred examples in the validation set, even a few evaluations of the validation set yields a maximum random baseline at least 10% higher than the standard random baseline. For example, a dataset of $n = 100$ examples has an expected max accuracy of 0.575 after only $t = 10$ evaluations. But it requires more than $t = 10,000$ evaluations to reach that expected max accuracy for a dataset with $n = 1,000$.

**Parameter extremes capture standard settings.** When $t = 1$ (only evaluating one random classifier), the expected maximum is simply the expectation of the binomial distribution. The maximum random baseline therefore subsumes the standard baseline as a special case. When $n$ is large, the expected maximum random accuracy is nearly the same as the standard random baseline because the binomial distribution is very concentrated around its expectation with large $n$. The implication of this result is that if an experiment has a large $t$, the standard random baseline is only a good indicator of random guessing if $n$ is also large. But while increasing $n$ is always a good idea theoretically, in practice it may not be feasible for the kind of diverse, fast-moving LM tasks that are the key use case for ICL.

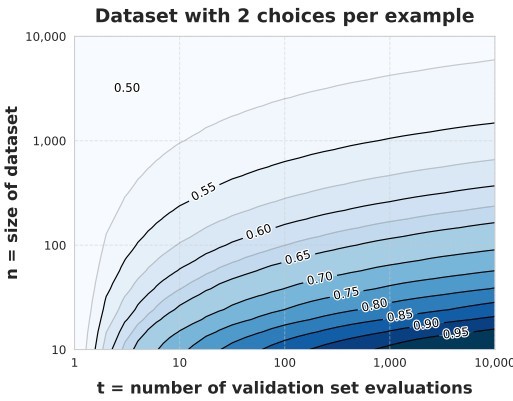

Figure 2: The expected maximum accuracy achieved among $t$ random classifiers on a binary classification dataset depends on $t$ and the size of the dataset.

## 5 Experiments

We evaluate the extent to which the maximum random baseline recontextualizes in-context learning performance. First, we study a deliberately simple and challenging setting of prompt demonstration selection using validation data: do heavily quantized language models outperform random baselines? We then move to a setting with a held-out dataset and find that comparing maximum validation accuracy to the maximum random baseline rather than the standard random baseline is more indicative of whether held-out accuracy exceeds random chance. Finally, we use the maximum random baseline to re-evaluate published results on prompt template selection and instruction selection.[4]

### 5.1 The standard random baseline overstates performance

Consider the task of choosing which demonstrations to include in a few-shot prompt. Prior work has shown that choosing different examples can lead to drastic differences in accuracy (Zhao et al., 2021). In this setting, we are going to report the accuracy of the prompt with the highest validation accuracy. Our goal is to get the best validation accuracy possible, and we have a budget of $t = 200$ different prompts to evaluate. Following Dodge et al. (2019), we report the expected maximum validation accuracy achieved by any prompt as a function of $t$, the number of evaluated prompts. We compare the best prompt's accuracy to both the standard random and maximum random baselines. See Appendix B for full details.

**Models and datasets.** We evaluate six LMs at the 7B-parameter scale: Llama-2-7b (Touvron et al., 2023), OLMo-7B (Groeneveld et al., 2024), Falcon-7b (Almazrouei et al., 2023), and their instruction-tuned counterparts: Alpaca-7b (Taori et al., 2023), OLMo-7B-Instruct, and Falcon-7b-instruct. We quantize the models to 4-bit for a particularly challenging setting (Dettmers et al., 2023). We use 16 BIG-bench Lite multiple choice tasks with their standard instruction templates (Srivastava et al., 2023). Our goal is to compare differences between standard and maximum baselines due to the nature of different few-shot tasks. Because we know analytically that the size of a validation set influences random baselines, we reduce size as a confounding factor by subsampling tasks to have a maximum size of 200 examples.[5] All baselines are calculated with respect to the size of the subsampled datasets.[6]

---

[4]Reproduction code is available at: `https://github.com/gyauney/stronger-random-baselines`

[5]Using smaller validation sets also results in substantial savings in computation when evaluating hundreds of example combinations, an important consideration for practitioners.

[6]For the seven datasets with $n < 200$, there are only $n$ possible demonstrations in the 1-shot setting. In these cases we compare to the proper maximum random baseline with $t = n$ instead of $t = 200$.

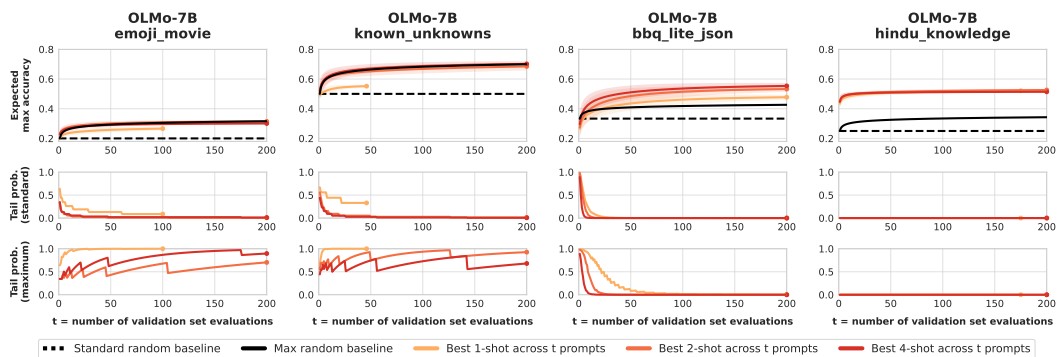

Figure 3: OLMo-7B 1-, 2-, and 4-shot beats the standard random baseline (dashed line) on four tasks in expected maximum validation accuracy. But accounting for validation set reuse with the maximum random baseline (solid black line), the best accuracies across prompts on the left two datasets are in fact no better than random.

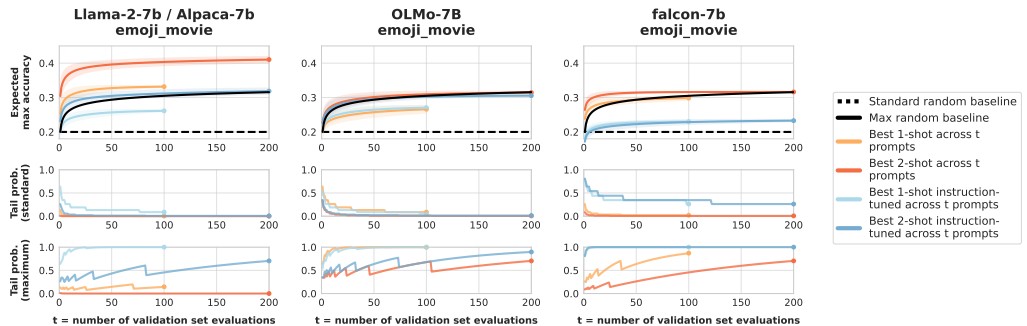

Figure 4: Expected maximum validation accuracy compared to the standard random and maximum random baseline for base and instruction-tuned models on a single hard dataset.

**Results for individual datasets.**    Figure 3 shows OLMo-7B performance for selected tasks. The $x$-axis shows the number of prompts $t$ evaluated on the validation set. As $t$ increases, the maximum random baseline also increases, sharply at first and then more gradually. For emoji_movie and known_unknowns, the expected maximum 1-, 2-, and 4-shot accuracies across multiple prompts are above the standard random baseline but *below* the maximum random baseline. To further illustrate how the baselines disagree, we can use their distributions to generate tail probabilities for the observed accuracies. In the middle row of panels, tail probabilities with respect to the standard random distribution are near-zero in almost all cases. The bottom row shows tail probabilities with respect to the maximum baseline: here 1-, 2-, and 4-shot all have high tail probabilities. Contextualizing performance against the maximum random baseline shows that these datasets are more challenging than they appear. Performance is high enough on tasks like bbq_lite_json and hindu_knowledge to be above random no matter which baseline is used. In these cases, tail probabilities for both baselines are near-zero. The maximum random baseline gives us additional confidence that performance is in fact good.

Figure 4 compares base and instruction-tuned models on the emoji_movie dataset. These results show that the standard random baseline substantially underestimates the difficulty of this dataset. Comparing instead to the maximum random baseline reveals that only Llama-2-7b outperforms random guessing. While the instruction-tuned falcon-7b models slightly outperform the standard random baseline, the standard tail probabilities show that a significant percentage of random classifiers still have higher accuracy.

**Aggregate results.**    Table 1 shows that out of 288 total experiments, maximum validation accuracy exceeded the standard random baseline in 255, but maximum validation accuracy

| BIG-bench Lite dataset | $n$ | Base model 1-shot L O F | 2-shot L O F | 4-shot L O F | Instruction-tuned 1-shot A O F | 2-shot A O F | 4-shot A O F | Total ◐ |
|---|---|---|---|---|---|---|---|---|
| novel_concepts | 32 | ●●● | ●●● | ●●● | ●●● | ●●● | ●●● | 0 |
| known_unknowns | 46 | ●◐○ | ●●● | ●●◐ | ○◐○ | ●◐○ | ●●◐ | 9 |
| code_line_description | 60 | ●●● | ●●● | ●●● | ●●● | ●●● | ●●● | 0 |
| emoji_movie | 100 | ●◐○ | ●◐○ | ●◐● | ○◐○ | ●◐○ | ●●◐ | 11 |
| conceptual_combinations | 103 | ●●● | ●●● | ●●● | ●●● | ●●● | ●●● | 0 |
| strange_stories | 174 | ●●● | ●●● | ○○○ | ●●● | ●●● | ●○○ | 0 |
| hindu_knowledge | 175 | ●●● | ●●● | ●●● | ●●● | ●●● | ●●● | 0 |
| bbq_lite_json | 200 | ●●● | ●●● | ●●● | ●●○ | ●●● | ●●● | 0 |
| formal_fallacies_syllogisms_negation | 200 | ○○◐ | ○◐○ | ○○○ | ○◐○ | ●◐○ | ○○○ | 12 |
| language_identification | 200 | ●◐○ | ●◐○ | ◐○◐ | ●◐● | ◐◐● | ○◐○ | 12 |
| logical_deduction | 200 | ●●● | ●●● | ●●◐ | ●●● | ●●● | ●◐◐ | 3 |
| play_dialog_same_or_different | 200 | ●●● | ○◐○ | ○○○ | ●●● | ●●● | ●○○ | 1 |
| strategyqa | 200 | ●●● | ●●● | ●●● | ●●● | ●●● | ●●● | 0 |
| symbol_interpretation | 200 | ○◐○ | ○○○ | ○○○ | ◐◐○ | ○○○ | ○○○ | 7 |
| vitaminc_fact_verification | 200 | ◐●● | ●●● | ○●○ | ●●● | ●●● | ●●○ | 1 |
| winowhy | 200 | ●●● | ●●● | ●●● | ●●● | ●●● | ●●● | 0 |
| Baseline disagreements per model ◐ | | 2  5  5 | 1  4  3 | 1  1  3 | 4  5  4 | 2  4  3 | 3  2  4 | |
| Total baseline disagreements ◐ | | 12 | 8 | 5 | 13 | 9 | 9 | 56 |
| Total percentage flipped ◐/(◐+●) | | 26% | 19% | 15% | 28% | 20% | 23% | 22% |

Table 1: Agreement between the standard random baseline and maximum random baseline when evaluating the maximum performance over $t = 200$ choices of prompt demonstrations on BIG-bench Lite. ○: best prompt performed worse than both baselines. ◐: best prompt performed **better** than the standard random baseline but **worse** than the maximum random baseline. ●: best prompt performed better than both baselines. $n$: number of examples. L: Llama-2-7b, O: OLMo-7B, F: Falcon-7B, A: Alpaca-7b.

exceeded the maximum random baseline in 199. The baselines disagreed in 56 experiments. This means that 22.0% of results that are above the standard baseline are not above the maximum baseline. Among base models, there are fewer baseline disagreements as the number of shots increases. Surprisingly, there are not fewer disagreements when moving from base models to their instruction-tuned counterparts. Specific datasets are responsible for many disagreements, the highest number of which come from formal_fallacies and language_identification. For many datasets, both baselines agree for all evaluations. Full per-dataset results are in Figures 9, 10, and 11 in Appendix E. Appendix C gives results for non-quantized models.

## 5.2 Maximum random baseline predicts held-out accuracy

Now that we have shown that the standard and maximum random baselines differ in many typical ICL settings, we turn to contextualizing held-out test set accuracy. Validation accuracy is often used as a first step to select a model, followed by reporting the selected model's performance on an additional held-out test set. But, of course, the more we look at the test set, the less useful it becomes. Here we show that the maximum random baseline is a better predictor of whether test performance will exceed random guessing than the standard random baseline and thus can avoid wasted test set evaluations.

**Setup.** Just as before, we choose the prompt with best performance on the validation data and report: the maximum validation accuracy, the standard random baseline, and the maximum random baseline. Given just the validation accuracy, we have to make a decision: is this prompt's test accuracy above or below random? Using the same models and 16 BIG-bench Lite tasks as above, we randomly partition each task into a validation set (75%) and a test set (25%). For a given model and task, we select the prompt from among $t = 200$ prompts with the highest validation accuracy. This prompt's validation accuracy is compared to the standard random and the maximum random baselines.

The prompt is said to outperform random on the corresponding held-out test set when its test accuracy is above the standard random baseline (because there has been no test set reuse). For each of 100 different splits of each dataset, we use validation accuracy to predict whether test set accuracy is above the standard baseline. We are comparing two "classifiers": standard predicts true when validation accuracy is above standard random, and max predicts true when validation accuracy is above the maximum random baseline. In both cases it is possible that a low quality prompt could have high validation accuracy due to random chance, so we do not expect perfect performance. Another reason we cannot expect perfect performance on this task is that just knowing that validation performance is above the standard random baseline means that performance is better than that of at least half of the random classifiers, not that it outperforms all of them. But we can evaluate which baseline gives us more insight into held-out accuracy.

**Results.** Evaluating all predictions across all splits of datasets, the held-out accuracy of the best prompt exceeds random 73% of the time. Figure 5 shows that when predicting whether test performance is above random from validation accuracy, standard achieves an accuracy of 0.82, AUROC = 0.67, and AUPR = 0.81. max has a lower accuracy of 0.79, but does better on AUROC = 0.80 and AUPR = 0.88. standard achieves higher accuracy on this task at the expense of many false positives—it often incorrectly predicts that test performance will be above random chance. In fact, the standard random baseline does only slightly better in this setting than simply predicting that all test accuracies will be above random. The maximum random baseline has higher precision—when validation accuracy surpasses the maximum random baseline we

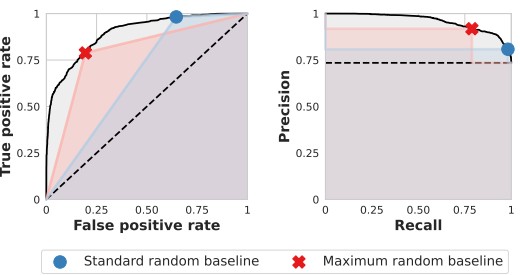

Figure 5: ROC and precision-recall curves when using maximum validation accuracy to predict whether held-out test accuracy will be above random chance. standard and maximum curves use binary predictions of above or below the given random baseline. The gray curve uses the distribution functions for confidence scores.

can be more confident that the corresponding held-out performance is above random. Results split by model and dataset are in Appendix D, as are different values of $t$ for validation set reuse. As the number of prompts $t$ that are being searched over increases, the maximum random baseline maintains a lower false positive rate even as the standard baseline's false positive rate increases (Figure 8, Appendix D).

We can additionally use each baseline's distribution function as a measure of how likely validation performance is to be above random. Instead of directly predicting above or below random as compared to each baseline's expected accuracy, we instead associate each point with the percentage of random classifiers that the validation accuracy is above. Both baselines produce the same rankings of likelihood above the baseline. This is because the distribution function of the maximum order statistic of the binomial distribution is a monotonic function of the distribution function of the binomial distribution (Equations 3, 6). Both baselines therefore yield the same ROC and precision-recall curves (AUROC = 0.90, AUPR = 0.96). The threshold for predicting above or below random performance is what changes between baselines, leading to the difference shown in Figure 5. In both cases, using the additional information provided by treating the random baselines as distributions contextualizes performance better than point estimates alone.

## 5.3 Choosing instructions and template formatting

The maximum random baseline can recontextualize existing results. Mizrahi et al. (2024) and Sclar et al. (2024) demonstrate that ICL performance is highly variable across instruction paraphrases and template formatting, respectively. As they release maximum validation accuracies and the number of prompts they evaluated, we are able to analyze the same experiments from the perspective of comparison to random baselines.

Mizrahi et al. (2024) report maximum validation accuracy across different instruction para-phrases for tasks from BIG-bench Lite and BIG-bench Hard. For BIG-bench Hard, they evaluate 11 models on 15 datasets and each time report the maximum validation accuracy across $t = 10$ prompts. Out of 165 experiments, 152 outperform the standard random baseline, and 147 outperform the maximum random baseline. 3.3% of results that exceed the standard random baseline do not exceed the maximum random baseline. For BIG-bench Lite, 9 models are evaluated on 13 datasets with 10 prompts each. Out of 117 experiments, 116 outperform the standard random baseline, and 109 outperform the maximum random baseline. 6.0% of results that exceed the standard random baseline do not exceed the maxi-mum random baseline. From the perspective of random baselines, these experiments are relatively robust because the models perform very well and $t$ is small.

Sclar et al. (2024) use Thompson sampling to evaluate $t = 320$ different prompt templates for Llama-2-70b on 53 different tasks from Super-NaturalInstructions (Wang et al., 2022) that were sampled to have $n = 1{,}000$ examples. Using the most generous parameter settings that make the maximum random baseline comparable to their setup, we find that 51 results of maximum accuracy across prompt templates are above the standard random baseline, and 46 are above the maximum random baseline. 9.8% of results that exceed the standard random baseline do not exceed the maximum random baseline.

Overall, the maximum random baseline can make us more confident when maximum valida-tion accuracies are above random performance while also identifying those model/dataset pairs with especially weak performance.

## 6 Discussion and conclusion

If an experiment reports the maximum performance across multiple evaluations on a dataset, the standard random baseline may significantly underestimate the probability of achieving that performance by random guessing. In-context learning is particularly susceptible to this danger because of its combination of small evaluation datasets, many prompt evaluations, and difficult tasks. In such settings, the expected maximum accuracy across multiple random classifiers is a stronger and more appropriate baseline. The baseline is easily calculated, and we release code for a drop-in replacement baseline.

Using the maximum random baseline for calibrating performance on validation sets can avoid inappropriate use of test sets. When a held-out test set is available, the maximum baseline can better predict the generalization performance of the best-performing prompt on the validation set when it is evaluated on a truly held-out test set. This can prevent test-set overuse: if the best prompt doesn't outperform the maximum random baseline on the validation set, then do not evaluate on the test set yet. The maximum baseline can gracefully account for dataset re-use in other ways, too. Even if an individual researcher strictly uses a benchmark's test set exactly once, benchmarks are re-used across studies (Koch et al., 2021). The maximum baseline can contextualize results on re-used test sets.

The maximum random baseline highlights the need to report more standard information to contextualize maximum accuracy across prompts. It is common to report validation size $n$, and reporting the number of prompt evaluations $t$ needed for this baseline is also good practice. Of a random sample of 20 papers from EMNLP 2023 that study ICL, six report using multiple prompts, three state that they report the maximum evaluation accuracy over multiple prompts, and only one of these reports how many times the validation set was used. We argue that reporting such parameters allows for proper contextualization of results and should be a best practice.

We reiterate the many calls for larger evaluation sets (Card et al., 2020; Bragg et al., 2021), this time to limit the variance of random guessing. But the reality of contemporary LM evaluation is that researchers have limited time and budget for computation and dataset development. Producing thousands of high-quality examples and evaluating many possible prompts on them may not always be feasible (Liang et al., 2023; Polo et al., 2024). The maximum random baseline proposed in this paper enables researchers to reduce validation size and therefore computation and cost, while still avoiding falsely positive prompt settings.

We leave it to future work to study the maximum random baseline across more models, tasks, prompt variations, and scoring strategies, as well as extensions to metrics beyond accuracy, such as F1. Future work can also study the rigorous specification of hypothesis tests for deciding whether generalization performance is likely to exceed random guessing, where dependence between classifiers poses a challenge for analysis. While we limit our analysis to in-context learning with language models, the stronger random baseline applies to any classification setting with evaluation set reuse.

Comparing a model's performance on a new task to random baselines is the first test of the model's capabilities. Random baselines remain relevant even as model performance increases across the board because increasingly difficult tasks are regularly constructed to probe the limits of performance. Ultimately, validation datasets will continue to be reused, especially given the variability of ICL performance. Baselines should account for this.

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

## A  Additional details of the expected maximum random baseline

**The probability of randomly guessing a correct answer does not depend on label balance.** Consider a random classifier $h$ that guesses labels uniformly at random, independently across examples. Given a labeled example $(x, y)$, the probability that this random guesser gets the correct label is $1/m$, where $m$ is the number of possible labels. It does not depend on the dataset having balanced proportions of labels:

$$
\begin{aligned}
P(h(x) = y) &= P\left(\bigcup_{\ell=0}^{m-1} \left(y = \ell \cap h(x) = \ell\right)\right) \\
&= \sum_{\ell=0}^{m-1} P\left(y = \ell \cap h(x) = \ell\right) \\
&= \sum_{\ell=0}^{m-1} P\left(y = \ell\right) P\left(h(x) = \ell\right) \\
&= \sum_{\ell=0}^{m-1} P\left(y = \ell\right) \frac{1}{m} \\
&= \frac{1}{m} \sum_{\ell=0}^{m-1} P\left(y = \ell\right) \\
&= \frac{1}{m}
\end{aligned}
$$

**Extending to tasks with different numbers of labels per example.** Some tasks allow each example to have a different number of possible labels. For example, the BIG-bench task `logic_grid_puzzle` has examples with 2 to 5 possible labels each. The number of correct guesses is modeled by a Poisson binomial distribution, where each independent trial can have a different probability of success, rather than a binomial distribution (Hong, 2013). The maximum order statistic of the Poisson binomial distribution can be found by plugging in its distribution function and probability mass function into equation 4. This extension is implemented in our code, though we leave further study to future work.

**Tail probabilities against random baselines.** The tail probability for the accuracy of a classifier $h_0$ against the standard random baseline is:

$$
\begin{aligned}
p_{\text{standard}} &= P(\text{acc}\,(h) \geq \text{acc}(h_0)) \\
&= 1 - P(\text{acc}\,(h) < \text{acc}(h_0)) \\
&= 1 - P(n\,\text{acc}\,(h) < n\,\text{acc}(h_0)) \\
&= 1 - P(X < n\,\text{acc}(h_0)) \\
&= 1 - \left(F(n\,\text{acc}(h_0)) - f(n\,\text{acc}(h_0))\right) \\
&= 1 - F(n\,\text{acc}(h_0) - 1)
\end{aligned}
$$

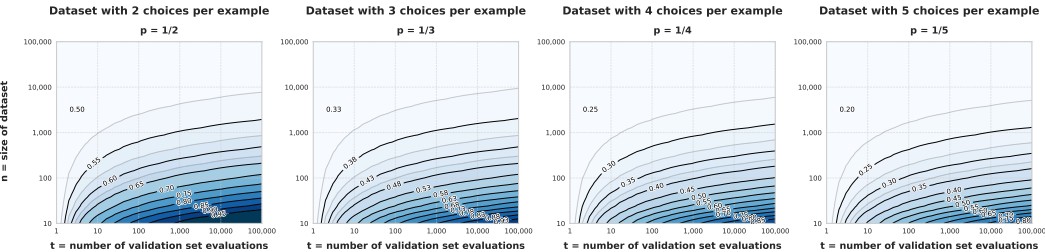

Figure 6: The expected maximum accuracy for various parameter settings. The leftmost plot is the same as Figure 2 but with larger parameter ranges.

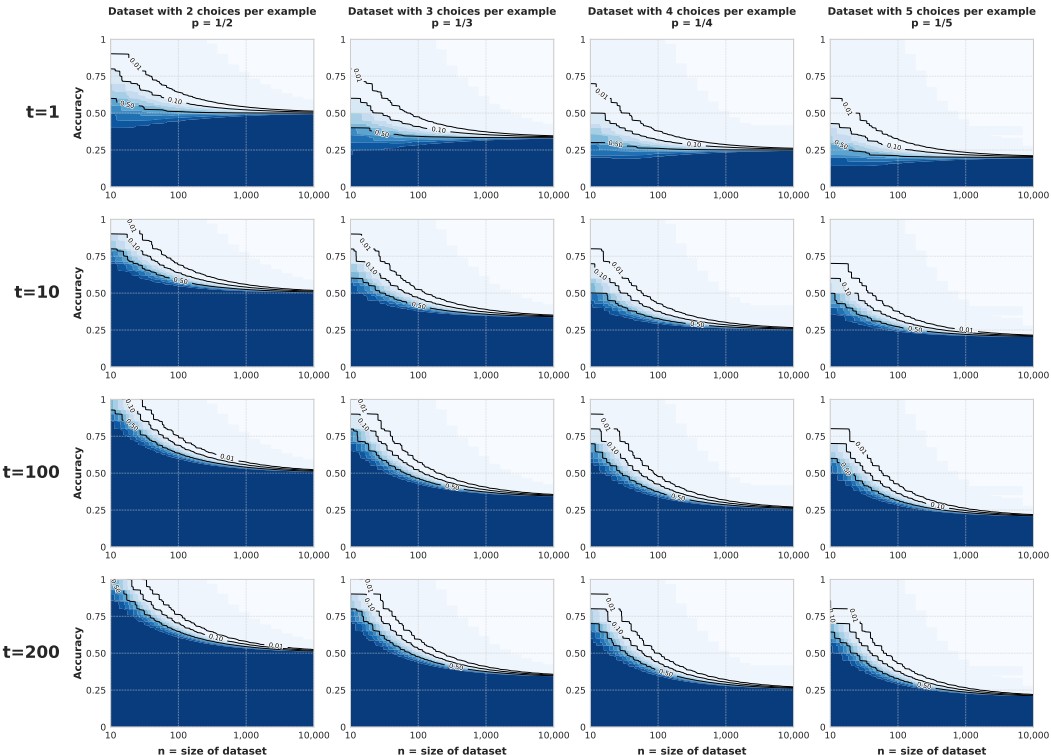

Figure 7: Tail probabilities with respect to the expected maximum random baseline for various accuracies and parameter settings. The top row ($t = 1$) corresponds to tail probabilities for the standard random baseline. Contours are shown at every 0.1 interval, with the 0.5, 0.1, and 0.01 contours labeled. Plots have been smoothed for ease of visualization.

The final line follows because the binomial is a discrete distribution over integers, so $F(k) = F(k - 1) + f(k)$ if we define $F(-1) = 0$. By similar reasoning, we get the tail probability with respect to the maximum random classifier:

$$
\begin{aligned}
p_{\max} &= P(\mathsf{acc}\,(h_{\max}) \geq \mathsf{acc}(h_0)) \\
&= 1 - P\Big(X_{(t)} < n\,\mathsf{acc}(h_0)\Big) \\
&= 1 - \big(F(n\,\mathsf{acc}(h_0) - 1)\big)^t
\end{aligned}
$$

**Additional plots.**   Figure 6 shows the expected maximum accuracy for various parameter settings of $n$, the number of examples in the evaluation dataset, $p$, the probability of guessing a correct answer on each example (one divided by the number of choices), and $t$, the number of random classifiers. Figure 7 gives tail probabilities for the expected maximum accuracy for various parameter settings. The top row ($t = 1$) corresponds to tail probabilities for the standard random baseline.

**Runtime.**   A naive implementation of the maximum random baseline depends mainly on $n$, the number of examples in the dataset. The most expensive settings that we consider still run in under 1 second for 10,000 examples.

# B   Evaluation details

We evaluate models in the in-context learning setting, where the model is supplied with a natural language prompt followed by demonstrations and a possible answer (Brown et al., 2020). This is commonly referred to as the few-shot setting, though *few-shot* can refer to other settings as well (Bragg et al., 2021).

**Datasets.** We use 16 datasets from BIG-bench Lite (Srivastava et al., 2023): `bbq_lite_json`, `code_line_description`, `conceptual_combinations`, `emoji_movie`, `formal_fallacies_syllogisms_negation`, `hindu_knowledge`, `known_unknowns`, `language_identification`, `logical_deduction`, `novel_concepts`, `play_dialog_same_or_different`, `strange_stories`, `strategyqa`, `symbol_interpretation`, `vitaminc_fact_verification`, `winowhy`. We use these because within a dataset, they have the same or very similar numbers of labels across examples. We begin with each full dataset, and for the tasks larger than $n = 200$ examples, we randomly sample 200 examples for evaluation and keep these fixed for all experiments. Lists of which examples were sampled can be found in the supplementary code.

**Prompts and demonstrations.**   A prompt consists of a natural language instruction with demonstrations in a given format. Each demonstration is an example along with its ground-truth label. In our deliberately simple setting, we keep the demonstrations constant across evaluation examples—a *prompt* refers to a fixed string with fixed demonstrations prepended to validation examples. We use each multiple choice task's standard instruction template provided with BIG-bench. We evaluate 200 different prompts for each combination of model, dataset, and number of demonstration shots. Demonstrations are chosen at random and appended to each other with \n. All prompts with their chosen demonstrations can be found in the code repository. For a given example with multiple choices, we select a model's answer as the label with the highest average log-likelihood per token (Holtzman et al., 2021).

For tasks with more than 200 examples, we sample 200 examples as a fixed set across all parameter combinations. We choose demonstrations from the remaining unsampled examples in the task. For the seven datasets with $n < 200$, there are only $n$ possible prompt demonstrations in the 1-shot setting. In these cases we compare to the proper maximum random baseline with $t = n$ instead of $t = 200$. For the same datasets, there is a subtlety regarding dataset size. Because the demonstrations come from the pool of available examples, the 1-shot, 2-shot, and 4-shot versions of the dataset differ by a few examples. Normally this would not make much of a difference, but the maximum random baseline depends on dataset size. For ease of visualization in Figures 3, 4, 9, 10, and 11, we plot the maximum random baseline for the dataset's full number of examples, which is slightly weaker than each setting's exact max random baseline (because the baseline increases with smaller $n$). All baseline judgments in Table 1 are with respect to each setting's exact maximum random baseline.

**Implementation details.**   All evaluations are implemented using NumPy (Harris et al., 2020) and Hugging Face `transformers` (Wolf et al., 2020). We use `bitsandbytes` to quantize models to NF4 4-bit with nested quantization and compute dtype bfloat16 (Dettmers et al., 2023). In cases where we compared quantized models to non-quantized models, performance was not consistently better or worse. We use an NVIDIA RTX A6000 with 48GB of RAM.

## C  Results for non-quantized models

The main paper evaluates the performance of quantized models. Table 2 gives full equivalent results for non-quantized models. In this setting, 15.8% of results that exceed the standard baseline do not exceed the stronger random baseline.

## D  Held-out set evaluations

For the results in Section 5.2, Table 3 shows per-model results. The maximum random baseline outperforms the standard baseline in AUROC and AUPR across all models on these datasets. Table 4 shows per-dataset results.

| | | Base model | | | Instruction-tuned | | | |
| | | 1-shot | 2-shot | 4-shot | 1-shot | 2-shot | 4-shot | |
| BIG-bench Lite dataset | $n$ | L O F | L O F | L O F | A O F | A O F | A O F | Total ◐ |
|---|---|---|---|---|---|---|---|---|
| novel_concepts | 32 | ●●● | ●●● | ●●● | ●●● | ●●● | ●●● | 0 |
| known_unknowns | 46 | ●◐● | ●●● | ●◐● | ●●◐ | ●●● | ●●◐ | 4 |
| code_line_description | 60 | ●●● | ●●● | ●●● | ●●● | ●●● | ●●● | 0 |
| emoji_movie | 100 | ●◐◐ | ●◐● | ●◐● | ●●◐ | ●●◐ | ●●◐ | 7 |
| conceptual_combinations | 103 | ●●● | ●●● | ●●● | ●●● | ●●● | ●●● | 0 |
| strange_stories | 174 | ●●● | ●●● | ○◐● | ●●● | ●●● | ●●○ | 1 |
| hindu_knowledge | 175 | ●●● | ●●● | ●●● | ●●● | ●●● | ●●● | 0 |
| bbq_lite_json | 200 | ●●◐ | ●●● | ●●● | ●●○ | ●●◐ | ●●● | 2 |
| formal_fallacies_syllogisms_negation | 200 | ◐◐◐ | ◐◐○ | ○○○ | ○◐◐ | ◐◐◐ | ◐○○ | 11 |
| language_identification | 200 | ●●● | ●●◐ | ◐○○ | ●●● | ●●◐ | ◐○◐ | 7 |
| logical_deduction | 200 | ●●● | ●●● | ●●● | ●●● | ●●● | ●●● | 0 |
| play_dialog_same_or_different | 200 | ●●● | ○○○ | ○○○ | ●●● | ●●◐ | ●○○ | 1 |
| strategyqa | 200 | ●●● | ●●● | ●●● | ●●● | ●●● | ●●● | 0 |
| symbol_interpretation | 200 | ○◐◐ | ○○○ | ○○○ | ◐◐◐ | ◐○○ | ◐○○ | 7 |
| vitaminc_fact_verification | 200 | ●●● | ●●● | ○●○ | ●●● | ●●● | ●●○ | 0 |
| winowhy | 200 | ●●● | ●●● | ●●● | ●●● | ●●● | ●●● | 0 |
| Baseline disagreements per model ◐ | | 1 4 5 | 1 2 1 | 1 3 1 | 1 2 4 | 2 1 5 | 3 0 3 | |
| Total baseline disagreements ◐ | | 10 | 4 | 5 | 7 | 8 | 6 | 40 |
| Total percentage flipped ◐/(◐+●) | | 21% | 10% | 14% | 15% | 17% | 15% | 16% |

Table 2: Non-quantized model results. Agreement between the standard random baseline and maximum random baseline when evaluating the maximum performance over $t = 200$ choices of prompt demonstrations on BIG-bench Lite. ○: best prompt performed worse than both baselines. ◐: best prompt performed **better** than the standard random baseline but **worse** than the maximum random baseline. ●: best prompt performed better than both baselines. $n$: number of examples. L: Llama-2-7b, O: OLMo-7B, F: Falcon-7B, A: Alpaca-7b.

| | Accuracy | | Precision | | Recall | | AUROC | | AUPR | |
| Model | Standard | Max | Standard | Max | Standard | Max | Standard | Max | Standard | Max |
|---|---|---|---|---|---|---|---|---|---|---|
| Llama-2-7b, Base model (77%) | **0.92** | 0.87 | 0.91 | **0.95** | **0.99** | 0.88 | 0.82 | **0.85** | 0.91 | **0.93** |
| OLMo-7B, Base model (72%) | **0.79** | 0.73 | 0.79 | **0.89** | **0.97** | 0.72 | 0.64 | **0.74** | 0.79 | **0.84** |
| Falcon-7b, Base model (67%) | **0.79** | 0.77 | 0.77 | **0.90** | **0.98** | 0.74 | 0.69 | **0.79** | 0.77 | **0.84** |
| Alpaca-7b, Instruction-tuned (84%) | 0.83 | **0.87** | 0.84 | **0.96** | **0.99** | 0.88 | 0.50 | **0.84** | 0.84 | **0.94** |
| OLMo-7B, Instruction-tuned (72%) | **0.78** | 0.72 | 0.78 | **0.88** | **0.97** | 0.71 | 0.63 | **0.73** | 0.78 | **0.83** |
| Falcon-7b, Instruction-tuned (68%) | 0.79 | **0.80** | 0.77 | **0.91** | **0.98** | 0.77 | 0.67 | **0.81** | 0.77 | **0.86** |
| **Total (73%)** | **0.82** | 0.79 | 0.81 | **0.92** | **0.98** | 0.79 | 0.67 | **0.80** | 0.81 | **0.88** |

Table 3: Per-model results when using whether maximum validation accuracy is above each baseline to predict whether held-out test accuracy will be above random chance. Percentages in parentheses indicate the proportion of trials where the best prompt's test accuracy was above random chance.

| BIG-bench Lite dataset | AUROC | | AUPR | |
|---|---|---|---|---|
| | Standard | Max | Standard | Max |
| code_line_description (100%) | **0.50** | **0.50** | **1.00** | **1.00** |
| bbq_lite_json (94%) | 0.67 | **0.78** | 0.96 | **0.97** |
| hindu_knowledge (100%) | **0.50** | **0.50** | **1.00** | **1.00** |
| novel_concepts (93%) | **0.50** | 0.46 | **0.93** | 0.92 |
| emoji_movie (62%) | 0.50 | **0.53** | 0.62 | **0.64** |
| vitaminc_fact_verification (83%) | **0.96** | 0.94 | **0.98** | **0.98** |
| conceptual_combinations (81%) | 0.50 | **0.55** | 0.81 | **0.83** |
| formal_fallacies_syllogisms_negation (37%) | **0.58** | 0.51 | **0.41** | 0.37 |
| known_unknowns (59%) | 0.49 | **0.57** | 0.59 | **0.63** |
| logical_deduction (81%) | **0.50** | 0.48 | **0.81** | 0.80 |
| play_dialog_same_or_different (59%) | **0.98** | 0.95 | **0.97** | 0.95 |
| strange_stories (69%) | 0.65 | **0.80** | 0.77 | **0.86** |
| symbol_interpretation (17%) | **0.78** | 0.50 | **0.33** | 0.17 |
| winowhy (100%) | n/a | n/a | **1.00** | **1.00** |
| strategyqa (93%) | **0.50** | **0.50** | **0.93** | **0.93** |
| **Total** (73%) | 0.67 | **0.80** | 0.81 | **0.88** |

Table 4: Per-dataset results when using validation accuracy to predict whether held-out test accuracy will be above random chance. Percentages in parentheses indicate the proportion of trials where the best prompt's test accuracy was above random chance.

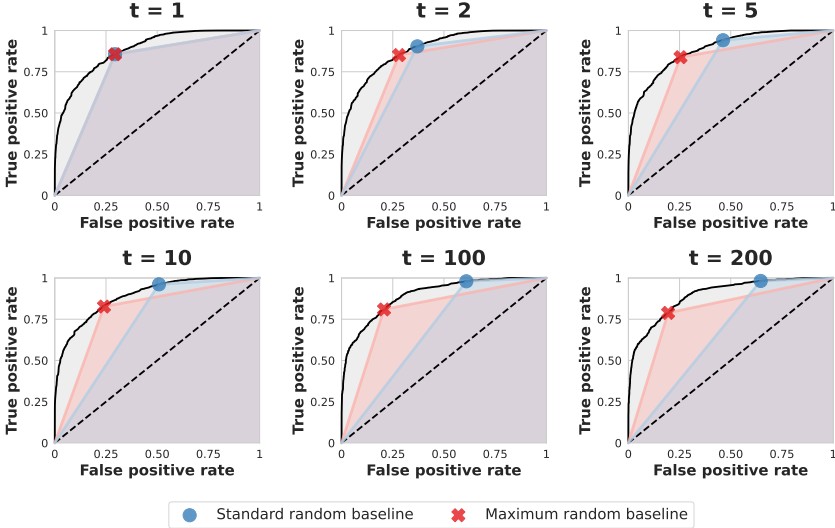

Figure 8: ROC curves for multiple values of $t$ when comparing maximum validation accuracy to baselines to predict whether held-out performance is above random guessing. The final panel is what is shown on the left side of Figure 5.

# E   Full results

This section provides full BIG-bench Lite results for Llama-2-7b and Alpaca-7b in Figure 9, OLMo-7B and OLMo-7B-Instruct in Figure 10, and Falcon-7b and Falcon-7b-instruct in Figure 11. We report expected maximum validation accuracy, as in Section 5.

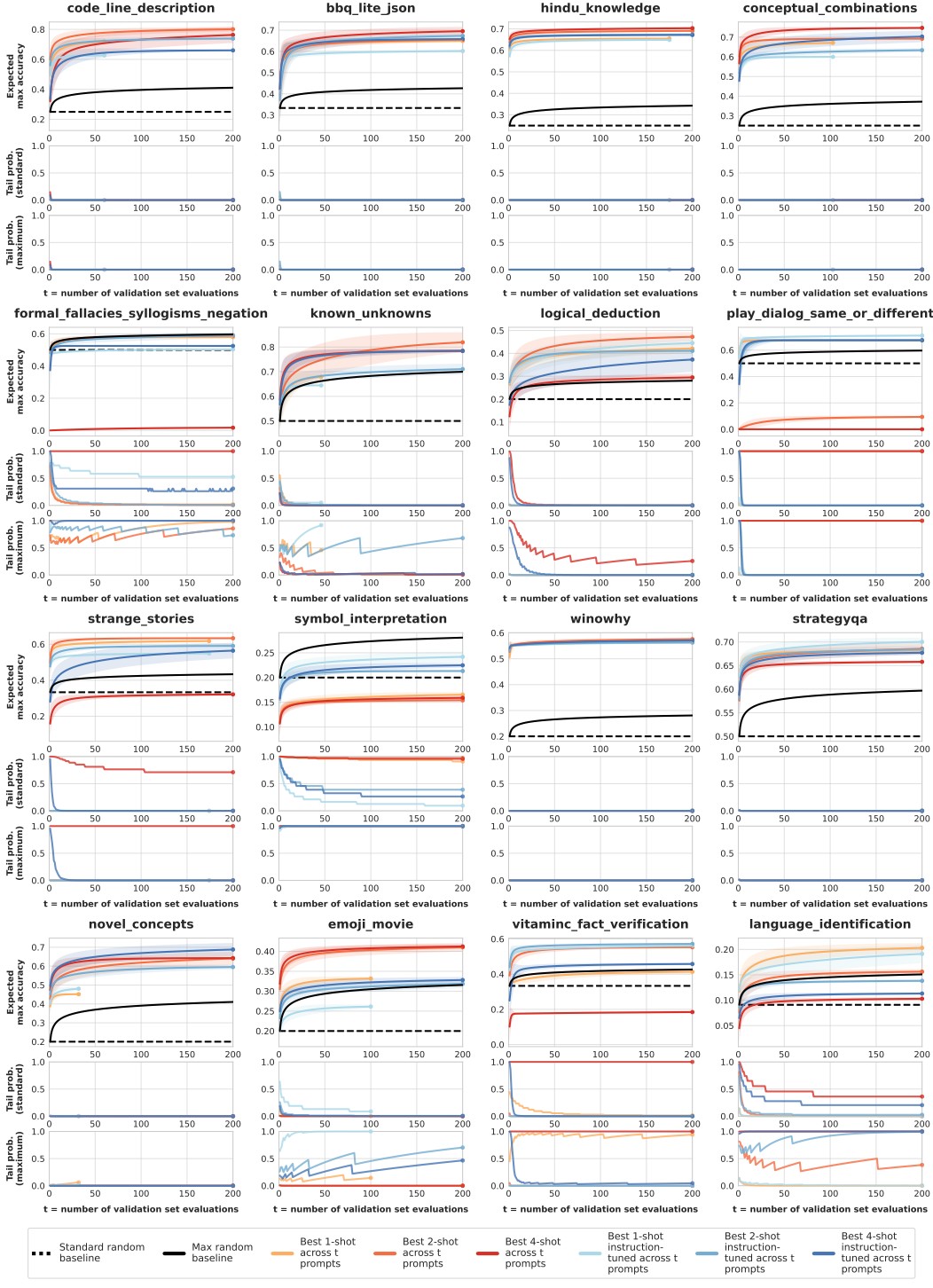

Figure 9: Full BIG-bench Lite results for Llama-2-7b and Alpaca-7b.

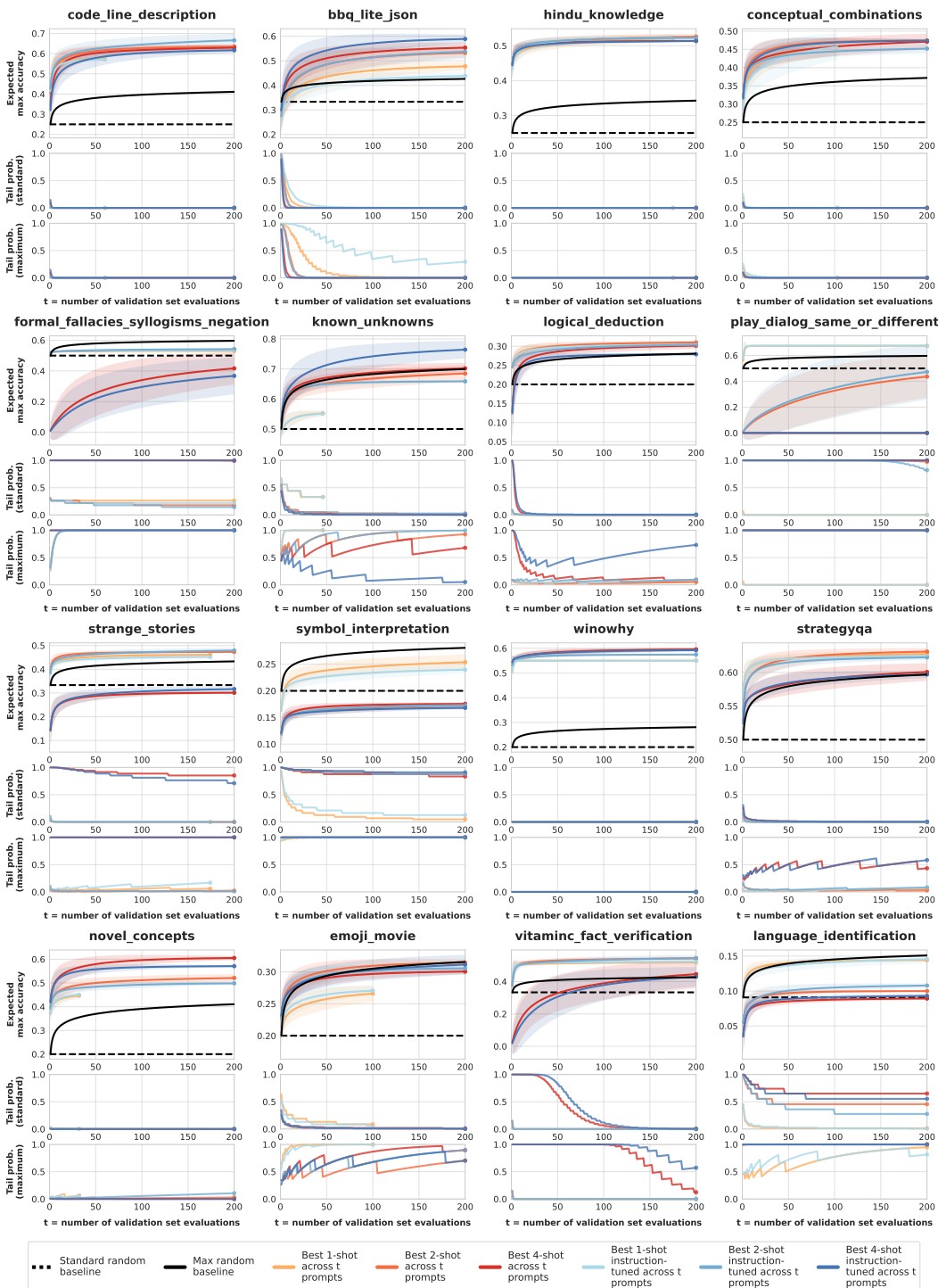

Figure 10: Full BIG-bench Lite results for OLMo-7B and OLMo-7B-Instruct.

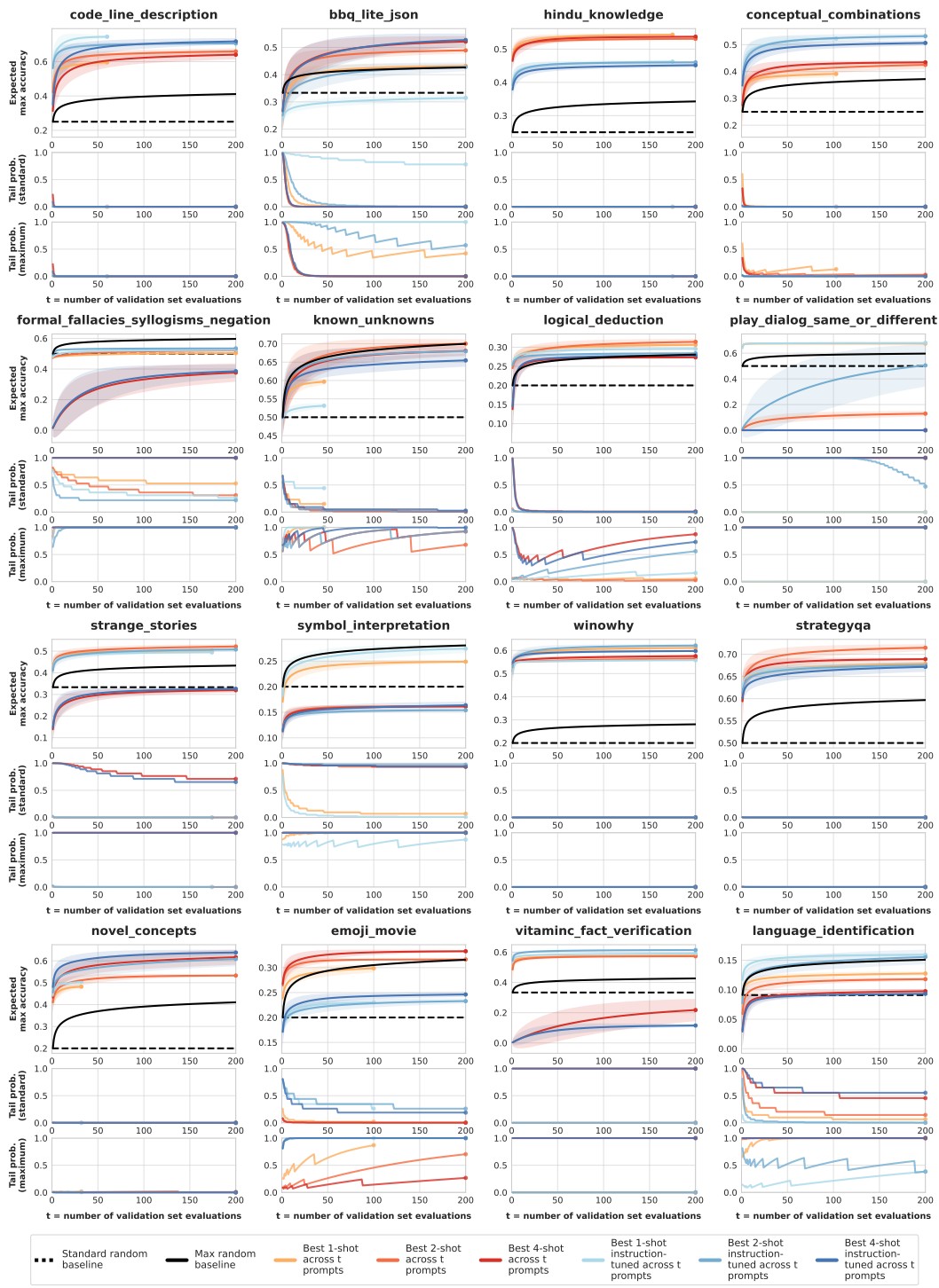

Figure 11: Full BIG-bench Lite results for Falcon-7b and Falcon-7b-instruct.

