# OpenReview forum: "Stronger Random Baselines for In-Context Learning"
_colmweb.org/COLM/2024/Conference — COLM_

### Official Review · Reviewer_pemZ · 2024-05-08

**Rating:** 6
**Confidence:** 4
**Ethics Flag:** 1

**Summary:**

The primary contribution of this paper is to propose a stronger random baseline when accounting for validation set reuse. The authors argue that the standard random baseline (expected accuracy of guessing labels randomly) is not sufficient when we allow for multiple evaluations on validation set, and when accounting for that (multiple evaluations on validation set and picking the max), the authors derive the closed form expression of the stronger random baseline, and show that many published results outperform the weaker standard random baseline but not the newly proposed baseline. The authors also argue that this baseline predicts held-out performance better.

**Reasons To Accept:**

- The new maximum random baseline is theoretically motivated and is intuitive to understand: if we allow multiple draws from the binomial distribution which the random classifier is doing, the expected value of the maximum accuracy will be higher than the accuracy if we only allow one draw. The baseline is built on this principle and I agree with the authors that this is also straightforward to implement and can be used as a drop-in replacement for the simple random baseline.
- The revelation that many published results do not, in fact, outperform this simple baseline is a potentially impactful finding.

**Reasons To Reject:**

One of my biggest concerns is as follows: the standard machine learning practice is *not* to rely on validation accuracy as the result to report. To give an example, if we train an image classifier, the standard practice is to evaluate the validation set at the end of each epoch, and then choosing the argmax for testing *once*. The test accuracy is typically what we care about, and we are only allowed to evaluate on it exactly once -- putting into the context of this paper, this means we are only allow to draw one realization of the binomial distribution (Eq (1)) once rather than $t$ times at *test-time*, which will reduce the proposed baseline back to the standard random baseline. Thus, even though on validation set, which is designed to be evaluated multiple times, the proposed baseline will be much stronger, it will make no difference at test time if standard protocol is followed. The authors seem to suggest that many reported papers do not properly use this elementary train-validation-test split and either report the validation accuracy only or evaluate on test sets multiple times -- while it is certainly not the authors' fault that basic data splitting protocol is not adhered to, I think that providing a stronger validation baseline seems to be the wrong question to answer when the real solution, which is not very difficult to implement and is virtually in any machine learning 101 course, is to use standard data splitting and to report test performance and allow for a single evaluation on test set.

As such, I felt that the contribution of this paper is somewhat conditional on a flawed practice of not properly using test set being widespread, and to my opinion using the test set properly as told in the textbook is a much better solution (even the authors have acknowledged so in Sec 6 *"While it is best practice to report test performance on a truly held-out set of examples"*). While I think it's unfair to harshly penalize the authors for violations of such basic rules by the others, I think it's necessary for this paper to strongly emphasize this caveat.

---

> ### Author Rebuttal · Authors · 2024-05-30
>
> Thanks for your thoughtful review! We really appreciate that your concerns get to the heart of our work. We think we can address most of them by reframing the intro/conclusion and emphasizing Section 5.2.
>
> ***Concern:** Standard practice is to select the best model using the val set and then evaluate that model on an unseen test set. The max random baseline makes no difference when train/val/test splits are properly used.*
>
> We agree that train/val/test splits are necessary and will state so in the intro. The max baseline should play exactly the same role as the standard baseline does now, where no one would bother evaluating on test if val is below a random baseline. Our work points out the fact that people are using the wrong random baseline here.
>
> Section 5.2 gives evidence that the max baseline on validation can—perhaps surprisingly—provide information on test accuracy. Comparing validation accuracy to the max random baseline is a better indicator than the standard baseline of whether the best classifier will outperform random guessing on an unseen test set. The test set should be used only once, and not wasted on classifiers that don't actually beat random guessing. We will reframe the intro/conclusion to a) emphasize Section 5.2’s findings and b) state that a test set should always be used for a good candidate model and that our goal is to avoid cases where you shouldn't even bother looking at a test set.
>
> ***Concern:** The max baseline is trying to correct a flawed practice that would be avoided by train/val/test splits.*
>
> We will state in the intro that the baseline is just one possible intervention on the widespread flawed practice of not using test sets properly. We will frame the intro and the results in Section 5.1 with the caveats you mention. We also hope that our work can trigger a community discussion about test set reuse across studies. Even when each researcher properly uses a test set only once, benchmark test sets are reused across studies to measure progress. When multiple models across different studies are compared on the same test set rather than redrawing fresh examples from the task distribution, the max baseline applies even to the test set.
>
> Please let us know if any of this addresses the core of your concerns. We are happy to discuss more!

---

> > ### Comment · Reviewer_pemZ · 2024-06-06
> >
> > Thank you for responding to my review. While I believe my concern that this paper won't be necessary if correct train/test split is adhered to still stands to some extent, 1) the authors mentioned they will emphasize the relevant parts of the discussions appropriately. As I mentioned in the original review, I do not think the authors should be penalized by widespread flawed practices in the community, and I agree that this work could possibly trigger discussions in the community; 2) the authors also highlighted two possible uses (that "no one would bother evaluating on test if val is below a random baseline" and the better predictive power of their validation metric on the test performance), which I think are worthwhile contributions even if train/test split is adhered to.
> >
> > As such, I am happy to increase the rating and I encourage the authors to incorporate all suggested changes into the final version of their paper.

---

> > > ### Author Response · Authors · 2024-06-06
> > >
> > > Thanks for being so open to discussion! We will incorporate all of the suggested changes into the paper, especially highlighting those two uses you mention. Your thoughtful feedback has improved our paper, and we really appreciate it.

---

### Official Review · Reviewer_nvye · 2024-05-09

**Rating:** 5
**Confidence:** 4
**Ethics Flag:** 1

**Summary:**

This paper introduces a novel random baseline for evaluating in-context learning performance.

In contrast to traditional random baselines that rely on prompts achieving the highest validation set performance,

the proposed method leverages multiple classifiers to generate more general and robust evaluation results.

Specifically, it first identifies the best-performing random classifier and then reports the expected performance derived from a variety of prompts.

The experiments demonstrate that this expected random baseline provides stronger and more reliable baselines compared to simple random baselines.

**Questions To Authors:**

Question 1: What do the classifiers mean exactly?
- As you mentioned, the model selection in prompt engineering could be different such as backbone models, input prompts, and hyperparameters.

Question 2: I would like to see the change of results according to the input prompt quality.

Suggestion
- I believe this concept could be developed further to create a general evaluation method or metric for prompt engineering.

**Reasons To Accept:**

- This paper addresses a critical issue in prompt engineering and in-context learning.
- Its motivation is simple and straightforward.
- The authors present extensive empirical results.
  - However, I could not understand how the empirical results support the claims.

**Reasons To Reject:**

- The usefulness of the proposed method is unclear.
  - It appears to simply replace a random baseline, offering no significant improvement.
- Additionally, with a small sample size (number of classifiers), t-tests and p-value comparisons may not be appropriate statistical tools.
- Finally, the computational cost seems significantly higher compared to a simple random baseline. The authors should address these concerns.

---

> ### Author Rebuttal · Authors · 2024-05-30
>
> Thanks for your review!
>
> ***Concern:** The usefulness of the proposed method is unclear. It replaces a random baseline, with no significant improvement.*
>
> We show that the standard random baseline is wrong, and that it systematically underestimates random guessing. In our experiments, >20% of "above random" results were actually below random. We present a more accurate replacement that may "outperform" seemingly strong results. Our goal is to better answer the question: “Does my model outperform random guessing on this task?” We help those who are developing new models, or choosing a prompt to deploy, not fool themselves into overestimating performance.
>
> ***Concern:** It is not clear how the empirical results support the claims.*
>
> Our main claim is that if you are choosing the best of multiple real classifiers, the correct baseline is the best of multiple random guessers, not the *average* random guesser. Our results show:
> 1. There is a meaningful difference between these baselines in real settings (Figures 1, 2, 3, 4; Table 1)
> 2. The stronger baseline better predicts held-out performance, reducing wasted test set evals (Figure 5)
> 3. The stronger baseline can re-evaluate published results (Section 5.3)
>
> ***Concern:** The computational cost is higher than the standard random baseline.*
>
> We will add to Section 4 that the max baseline runtime mainly depends on the number of examples, and at its most expensive still runs in under 1 second for 10,000 examples.
>
> ***Concern:** t-tests and p-values may not be valid for small sample sizes.*
>
> We do not conduct any statistical hypothesis tests, such as t-tests. $t$ is a variable for the number of random guessers. We use “p-values” simply to summarize how many random guessers a given model outperforms. We will rename “p-value” to “tail probability” to avoid confusion.
>
> ***Question 1:** What do the classifiers mean?*
>
> We will clarify what “classifiers” means depending on context. The max baseline reports the expected maximum accuracy across uniform random guessers. In Section 5.1 and 5.2, classifiers correspond to prompts with different few-shot demonstrations. In Section 5.3, classifiers are different prompt templates and instruction paraphrases.
>
> ***Question 2:** How do results change with input prompt quality?*
>
> Section 5.3 analyzes previously published results that vary the quality of prompt templates and prompt phrasing, finding that the standard and max baselines can disagree there as well. We will expand this.

---

> > ### Comment · Reviewer_nvye · 2024-06-06
> >
> > Thanks for addressing my comments.
> >
> > Despite of the author response, I will maintain my current scores.
> >
> > It does not mean this work is not valuable, but I personally think that this work needs minor revisions or can be much improved, as the authors said.

---

> > > ### Author Response · Authors · 2024-06-07
> > >
> > > Thanks for reading our response. We appreciate that you would like to see a revision, though we believe that we have addressed your concerns both in the original paper and in our response. We're uncertain what additional specific changes would satisfy these concerns.

---

### Official Review · Reviewer_WAy5 · 2024-05-17

**Rating:** 6
**Confidence:** 4
**Ethics Flag:** 1

**Summary:**

This paper proposes a well-justified stronger random baseline for in-context learning (ICL), presenting the simplest case-study for the new random baseline: ICL in the context of single-label classification tasks. The idea behind the baseline is taking the expected best performance across multiple random classifiers, where the actual strength of the baseline naturally depends on the choice of the population/validation set, as well as on the number of the evaluated random classifiers. These dependencies are also analysed in the paper with some practical tips provided to the researchers and practitioners.

The importance of having such a more elaborate baseline especially for ICL-style experiments is nicely motivated and the post-hoc analysis on BIG-Bench Lite reveals that the choice of a different random baseline can even have large impact on how one perceives performance of the actual method/system.

**Questions To Authors:**

This is not a question, but rather a suggestion. It would be useful to maybe provide a table or a simple plot that shows how expected maximum accuracy changes with respect to its crucial parameters (n and t) so that the reader gets a better overview of these dependencies.

**Reasons To Accept:**

- The main idea is nicely motivated and elaborated on from multiple perspectives: (a) practical need; (b) theoretical justification, and (c) empirical findings.
- The analyses delve deep into the main problems of the standard random baseline for ICL and how different choices of random baselines can provide different experimental conclusions.
- Some practical tips have been provided, and the paper provides a nice balance between more theoretical insights and its downstream/practical value.
- The paper is very well written and the main concepts and ideas have been well described and I enjoyed reading the paper.

**Reasons To Reject:**

- The choice of tasks could have been made more comprehensive so that the reader also gets a perspective related to task complexity, number of labels, and some calibration issues and properties of ICL (e.g., see https://arxiv.org/pdf/2309.17249 for some work on calibration for ICL)
- The paper leaves too many directions for future work which it should have ideally explored already. For instance, at least some directions should have been investigated among the ones listed in Section 6:
1. more models
2. more tasks
3. prompt variations
4. scoring strategies.
5. other evaluation metrics
6. classification tasks with different properties (single-label vs multi-label or a small number of labels vs a larger number of labels).
Put simply, more experiments would have strengthened the main findings of the paper and possibly the wider adoption of the proposed stronger random baseline.

- I am also unsure if the proposal for the stronger random baseline holds in a standard setup where we do selection/optimisation on validation set multiple times and then just port a single best model to the test only once. The random baseline on the test set is then again the standard random baseline? I might be missing something here, so I would definitely like to see this discussed.

---

> ### Author Rebuttal · Authors · 2024-05-30
>
> Thanks for your review!
>
> ***Suggestion:** Add a plot showing how expected max accuracy varies with n and t.*
>
> We will clarify that Figure 2 at the top of page 5 is this plot! Figure 6 on page 13 shows the same for 3-/4-/5-way classification.
>
> ***Concern:** Does the max random baseline hold when evaluating a selected model on an unseen test set?*
>
> Thank you for asking this. You are exactly right about the standard baseline on the test set if you only evaluate one classifier. Section 5.2 shows that the max random baseline can—perhaps surprisingly—predict performance on an unseen test set. Comparing validation accuracy to the max random baseline is a better indicator than the standard baseline of whether the selected classifier will outperform random guessing on an unseen test set. We will expand Section 5.2 and emphasize these findings in the intro.
>
> ***Concern:** Many directions for future work (more models/tasks/prompts/scoring strategies/metrics, number of labels, task complexity, calibration) should have been explored already.*
>
> We agree that it’s always more convincing to have more experimental settings. Do you have specific concerns that additional experiments could answer beyond showing general robustness?
>
> We think discussing the variety within the experiments we have already run will address some of your concerns:
> - Task complexity, number of labels: We study 6 models on 16 tasks in 3 few-shot settings, with numbers of labels ranging from 2 to 11 labels. The max baseline works well for all tested label numbers. The baseline matters most for hard tasks where models are right at the cusp of beating random guessers.
> - More models, tasks, prompts: Section 5.3 uses published results to analyze the max random baseline for 1) prompt template choice on Llama-2-70b accuracy across 53 tasks from Super-NaturalInstructions, and 2) instruction paraphrasing across 11 models on 15 BIG-bench Hard tasks and 9 models on 13 BIG-bench Lite tasks.
>
> As a comparison, we take the Batch Calibration paper you linked as a model of good practice. It studies 1 new method (and 4 baselines) across 3 models on 13 tasks and 1 model on 6 more tasks.
>
> Our goal with this paper is to 1) introduce the max random baseline and 2) demonstrate that the baseline makes a difference in important settings. Experiments in future work will add nuance to the max baseline’s impact, but we don’t think any outcome of those experiments would change the correctness or usefulness of the new baseline.

---

> > ### Comment · Reviewer_WAy5 · 2024-06-04
> > **After reading the response...**
> >
> > ...and the responses to the other reviewers, I conditionally adjust my score, where the condition is to really integrate the new clarifications and motivations into the main paper. Many thanks for taking the time to provide a well-written response!

---

> > > ### Author Response · Authors · 2024-06-06
> > >
> > > Thanks for being so open to discussion! We will absolutely incorporate the clarifications and motivations into our paper--we really appreciate your feedback.
> > >
> > > (We also want to quickly note that we're not seeing any adjusted score on our side--if that's intentional in order to show that the score update is conditional, then no worries and please ignore this.)

---

### Official Review · Reviewer_fFwf · 2024-05-18

**Rating:** 7
**Confidence:** 3
**Ethics Flag:** 1

**Summary:**

This work proposes an alternative approach to LLM in-context learning evaluation against a random baseline by considering the performance of the best random baseline across multiple runs. The proposed method is shown to reframe the performance of several state-of-the-art large models on a subset of BigBench tasks, serve as a better predictor of test set accuracy, and provide more precise information about multi-prompt evaluation results.

**Questions To Authors:**

1. Have you empirically evaluated the impact of class imbalance on the performance of a baseline that samples classes uniformly?

**Reasons To Accept:**

1. The problem of robust evaluation studied in the paper is highly important, especially in light of other advances in the field
2. The proposed method is simple to understand and implement, yet provides a theoretically grounded way of estimating maximum performance across classifiers
3. The set of experiments is sufficiently extensive to demonstrate the benefits of the method in several settings

**Reasons To Reject:**

1. I am slightly concerned about the potential impact of the method, because in a significant amount of research, language models are compared to each other and not to a random baseline. The maximum random baseline might indeed outperform the accuracy of state-of-the-art models on some tasks, but often the point of comparison is to measure progress in capabilities on a certain task and not to check if the model is meaningfully better than random choice. For instance, p-values against random baselines might see limited applicability because evaluations frequently involve multiple models and therefore should be viewed as multiple comparisons from a statistical perspective.
2. The approach appears to assume that the ground truth distribution of class labels is uniform: unless I misunderstand something, the probability of the random classifier being correct is not $1/m$ if a label is chosen at random. I wonder if the proposed approach can be extended to accommodate for such cases; at the very least, it would be worth mentioning the potential class imbalance when describing the method

---

> ### Author Rebuttal · Authors · 2024-05-30
>
> Thanks for your review!
>
> ***Concern:** Models are more often compared to each other rather than to random baselines.*
>
> We agree that researchers usually compare to other models. We're saying that they should also make sure they beat a more honest random baseline. Our results show that when datasets are small, even models that perform well above the standard baseline can be beaten by our new baseline. In such cases, it doesn’t matter as much which model is better if none outperform random guessing. We will also emphasize that our goal is not just to support academic papers, but to support practitioners.
>
> ***Concern:** The discussion of random baselines assumes that labels are uniformly distributed between all classes. When labels are imbalanced, the probability of a correct guess is not 1/m.*
>
> Thank you for encouraging us to look at this! We will add the following to Section 3.
>
> Surprisingly (even to us!), the probability that a uniform random guesser gets the correct label is $1/m$ and does not depend on class balance. The probability that a uniform guesser is correct on an example in a binary task (0/1 labels) where a possibly imbalanced $q$ fraction of examples are labeled class 0 does not depend on $q$. Let $y$ be the correct label and $g$ be the guess:
> $$P[y = g] = P[(y = 0 \cap g = 0) \cup (y = 1 \cap g = 1)] =  P[y = 0]P[g = 0] + P[y = 1]P[g = 1] = q * 0.5 + (1-q) * 0.5 = (q + 1 - q) * 0.5 = 0.5$$
>
> When a dataset is imbalanced, a classifier that guesses the majority label will have a higher accuracy than a uniform random guesser. But classifiers that always guess the majority are not randomized, so taking the max of multiple will not increase accuracy. Random guessing proportional to the class imbalance will do better than uniform guessing: the probability of a correct guess becomes the sum of squared label probabilities (which is a constant > 1/m), so we can reuse the max baseline framework with a success probability p > 1/m. We’ll expand this analysis—sorry this is just a sketch due to word limit.
>
>
> ***Concern:** p-values comparing to random baselines are not applicable in a setting with multiple models because of multiple comparisons.*
>
> We are using p-values as a way to summarize how many uniform random guessers a given model outperforms. We don’t use them for any hypothesis testing, so we think the multiple comparisons problem should not apply here. We will rename “p-value” to “tail probability” to prevent confusion.

---

> > ### Comment · Reviewer_fFwf · 2024-06-05
> >
> > Thank you for the detailed response! I am satisfied by the responses, and I have updated my score in light of the discussion, assuming authors will make the described changes to the camera-ready version.
> >
> > To be clear, I believe the paper should be accepted, but I am still not sure about its potential impact, because I struggle to envision truly broad adoption of this baseline in future ICL works. For example, when the core contribution of a study is to improve upon previous capabilities of existing models via new ICL methods, it is probably more important to compare metrics relative to the original model and standard ICL prompting instead of referring to the random baseline in absolute terms. Still, I appreciate that this work aims to improve how the performance of LLMs on benchmarks is perceived, as the random baseline provides a more balanced perspective on the model capabilities.

---

> > > ### Author Response · Authors · 2024-06-06
> > >
> > > Thanks for reading our response! We appreciate your additional feedback, and we will keep your concerns in mind as we edit the paper.

---

### Decision · Program_Chairs · 2024-07-10

**Decision:**

Accept

**Comment:**

The paper proposes an alternative random guessing baseline that sets a stronger bar for meaningful in-context learning (ICL). This new baseline prompts rethinking of the success of current ICL methods, as they do not outperform this stronger baseline on more than 20% of the datasets. Moreover, when considering accuracy on a truly held-out set, this proposed random baseline still outperforms the standard random baseline.

Strengths
* The problem of robust evaluation is crucial, and the paper offers fresh insights into this topic (Reviewer fFwf, WAy5, nvye, pemZ).
* The proposed baseline is very simple and is theoretically grounded (Reviewer fFwf, nvye, pemZ).
* The paper is well-written, with the theoretical justification and practice values (Reviewer WAy5).

Weaknesses
* The main concern is that the standard ML practice does not rely on validation accuracy, so the paper is arguable based on a flawed practice (Reviewer pemZ). Section 5.2 provides evidence that the method generalizes to the held-out test set, but the comparison is made against a basic random guessing baseline rather than ICL methods. It remains unclear how the proposed random baseline would perform compared to current ICL methods if the setting in Section 5.2 were the default.

Diverging opinions
* Experiments are extensive in some dimensions (Reviewer fFwf, nvye) but are not in others (WAy5). The paper includes many datasets and many models which is a plus, but is missing prompt variations and scoring strategies. In addition, the conclusion of the paper mainly holds with 1-shot and 2-shot but less with 4-shot, and less likely with more than 4 (which is a typical ICL setting).